# TimeChain: A Secure and Decentralized Off-chain Storage System for IoT Time Series Data

## ABSTRACT

Blockchain-based distributed storage systems offer enhanced security, transparency, and lower costs compared to traditional centralized storage, making them ideal for peer-to-peer collaboration. However, with the trend towards the Web of Things (WoT), lower transaction speeds and higher computational requirements limit their access to high-density data such as IoT. To address this, we propose TimeChain, an efficient off-chain blockchain storage system for IoT time series data. TimeChain batches discrete time series data, storing only the hash value of each batch on-chain while keeping the complete data off-chain. This significantly reduces storage overhead on the blockchain and storage latency by 37.4 times. In order to reduce the additional transmission latency in range queries, TimeChain employs an adaptive packaging mechanism. We convert the batching problem to a graph partitioning problem by representing data and historical co-query as graph vertices and edge weights respectively. To reduce the size of the transmission size in data integrity verification, a Locality-Sensitive Hashing (LSH)-based data integrity verification mechanism, which minimizes the data required for integrity checks by transmitting only non-redundant parts. TimeChain also integrates a node selection mechanism based on consensus protocol, which reduces the overhead by combining node selection and consensus processes. Our evaluation shows a reduction in query latency by 64.6% and storage latency by 35.3% compared to existing systems.

## KEYWORDS

IoT Series Data, Blockchain, Database

## 1 INTRODUCTION

The Web of Things (WoT) is an important trend led by W3C that aims to address Internet of Things (IoT) interoperability issues by adopting the proven technologies and patterns of the Web [26]. According to Gartner's analysis, billions of deployed IoT devices in the future will generate zettabytes (ZB) of data [19]. With the trend towards WoT, this ZB-level data needs to be connected to the web. For such large-scale data, the use of decentralized servers (e.g., AWS IoT[2], Aliyun Cloud [1]) to manage it suffers from problems such as a single point of failure [15]. Although distributed databases can avoid a single point of failure, they are susceptible to data tampering attacks in scenarios that require a high degree of transparency, such as IoT data sharing [10], due to weak data security and non-tamperability [29].

Blockchain, characterized by traceability and immutability, can solve the single point of failure problem of centralised storage and the malicious tampering problem of distributed databases [18]. It stores data in a decentralized ledger and uses consensus protocols

to resolve conflicts between equal nodes, which way has enhanced security, increased transparency, and lower costs, supporting peer-to-peer collaboration [30]. Though its great potential, its lower transaction speed and higher computational requirements make it only widely used in areas with huge value but very low density, e.g., financial services, supply chain management, etc.

There are many researchers working on improving the performance of blockchain-based storage systems. Based on the storage location of data, these works can be divided into two classes, namely on-chain storage and off-chain storage. For on-chain storage, data is included as part of the transactional records stored on the blockchain and users acquire these data by the index (i.e., Merkle Patricia Trie, MPT). Existing work improves system usability by providing user-friendly query language [33, 36, 46] and system throughput by improving indexing scheme [25, 43], blockchain storage sharding [14, 18, 42]. However, the overhead of storing IoT data on-chain is very high. For large-volume and fast-generating IoT data, storing it continuously on the blockchain requires the processes of achieving consensus and ledger replication, which can lead to significant storage pressure and communication overheads. Therefore, a more practical approach for IoT data storage is to leverage off-chain storage solutions. For off-chain storage, data are stored outside of the blockchain, and the blockchain stores only the necessary metadata or references to the data, such as hashes or cryptographic pointers. Off-chain storage offers greater scalability and lower cost than on-chain solutions, so there is a lot of excitement in industry and academia about off-chain storage, e.g. Storj [22], BigchainDB [27], Sia [4], etc. However, existing works are mostly designed for the storage of large files. For small-size IoT data, considering the massive amounts of IoT data, storing a hash of each data item in the blockchain would incur incredible overhead. Besides, IoT application scenario often requires storage system support for efficient queries (e.g., aggregation queries), which is not supported by existing file-based storage systems.

In this paper, we propose TimeChain, an efficient off-chain blockchain storage system for time series data. This system batches discrete time series data, stores only the hash value of each batch on the chain, and keeps the complete original data off-chain. This batch storage method significantly reduces data overhead. We conduct a measurement on the performance of the off-chain blockchain storage system. According to our measurement, compared to single data storage, the storage latency is reduced by an average of 37.4 times. This storage performance makes blockchain-based storage for time series data feasible.

However, it is undeniable that this also impacts query performance. Specifically, when a user executes an aggregate query, inefficient batch processing methods result in fetching data across multiple transmission nodes, causing additional transmission delays. In addition, since only the hash values of batches are stored on the blockchain, the storage system must transmit additional information (e.g., the hash path of a Merkle tree) to support the

*WWW'25, April 28–May 2, 2025, Australia*
2024. ACM ISBN 978-x-xxxx-xxxx-x/YY/MM
https://doi.org/10.1145/nnnnnnn.nnnnnnn

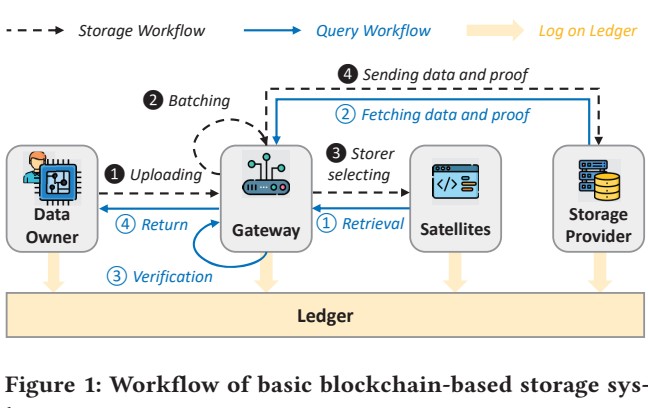

**Figure 1: Workflow of basic blockchain-based storage systems.**

data owner in performing data integrity verification. This further increases the query overhead.

In order to reduce the number of storage nodes spanned during range queries, we propose a novel adaptive packaging mechanism. We transform the batch processing problem into a graph partitioning problem by building an undirected weighted graph from the raw data. We use the spectral clustering algorithm to solve the partitioning problem by grouping frequently queried data together to reduce the number of node accesses during aggregated queries.

To further reduce query latency while ensuring system security, we propose a consensus-based storage node selection mechanism. We consider storage node reputation and transmission distance together when selecting nodes. In order to reach the node selection decision quickly in blockchain-based storage system, we combine the consensus process with nodes to reduce the propagation delay.

To reduce the transmission overhead during verification, we propose a data integrity verification mechanism based on location-sensitive hashing (LSH). This mechanism transmits only the non-redundant portion of the proof based on the similarity between neighboring time series data points, thus significantly reducing the data required for integrity verification.

We implement TimeChain based on top of production-ready open-source components such as Hyperledger Fabric and IPFS, and evaluate the performance of TimeChain. The result shows that compared to existing blockchain-based storage systems, TimeChain reduces 64.6% query latency and 35.3% storage latency on average.

## 2 BACKGROUND AND PRELIMINARY STUDY

To improve the performance of blockchain-based distributed databases, we propose a basic off-chain storage system and conduct a measurements study on it.

### 2.1 Blockchain-based Storage System

As shown in Fig. 1, a basic blockchain-based distributed storage system has four main roles, namely data owners, gateways, satellites, and storage providers. Data owners request storage resources and query data. Gateways provide an interface for consumers to interact with the network, allowing them to upload, download, and manage their data. Satellites coordinate the communication between owners and providers. They provide file auditing (Audit) or POR (retrievability) related functions and storage payment processing.

To ensure the security of the process, satellites are often operated in the form of smart contracts. Storage providers store and retrieve data to earn rewards by providing storage and bandwidth resources. For the storage provider to quickly provide proof of integrity to the data owner with flexible queries, the proof data needs to be stored in the storage provider as well. The storage provider's service information such as remaining storage space will be recorded in the distributed ledger along with the interaction records to ensure security. Generally, the data storage and query procedure can be summarized as follows:

**Data Storage:** ❶ *Uploading*: The data owner uploads data through a gateway interface. ❷ *Batching*: The gateway batches the time-series data and generates data integrity proofs of each batch. ❸ *Storer selecting*: The satellites help the gateway in discovering the optimal storage node for storing the data. ❹ *Sending data and proof*: The raw sensor data and the integrity proofs are sent to the optimal storage provider. The metadata of the data batch are recorded in the distributed ledger.

**Data Query:** ① *Retrieval*: The data owner requests to download their data, and the gateway interacts with the satellites to retrieve the location of the corresponding storage provider. ② *Fetching data and proof*: The gateway fetches data and integrity proofs from storage providers. ③ *Verification*: The gateway verifies the integrity of the downloaded data by checking data integrity proofs. ④ *Return*: The gateway returns the data to the data owner.

### 2.2 Measurement Study

In this section, we conduct a preliminary study to evaluate the performance of the basic blockchain-based storage system. We implement the storage system based on Hyperledger Fabric [3]. This test network consists of 5 nodes, with 1 node as both gateway and 4 nodes as satellites. We simulate 300 storage providers around the world [13, 47].

**Storage Performance.** We set the data owner to generate 20 packets of 56 bytes per second and store them within 20 seconds. The storage performance results are shown in Fig. 2a. Batch storage reduces latency by about 37.4 times compared to storing each data individually. This is mainly due to the fact that the larger batch size reduces the number of on-chain transactions.

**Query Performance.** We then test the performance of the range query, as shown in Fig. 2b. Unfortunately, the results show that the query performance of batch storage solution is relatively poor, with an average latency of 165.4ms under different batch sizes, which cannot meet the needs of many IoT scenarios. For example, the latency of an autonomous driving application is less than 50ms [9], and the latency of earthquake monitoring is less than 100ms [8].

### 2.3 Root Causes of Bad Query Performance

To find out the cause of the poor query performance, we conduct an in-depth investigation and summarize the causes into the following three points:

**1) Multiple Queried Spanning Batches.** Queries for time series data often encompass multiple data points, such as range queries, aggregation queries, filter queries, and so forth. A single query may span multiple batches if not packaged appropriately. We evaluate the number of batches spanned by each query using

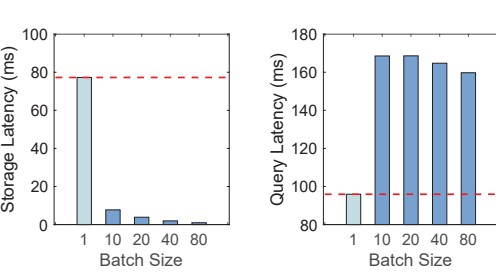
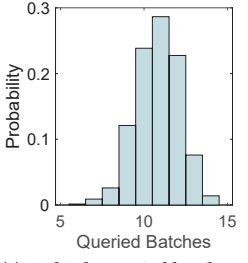

**(a) Storage latency.** **(b) Query latency.**

**Figure 2: Performance of the basic system.**

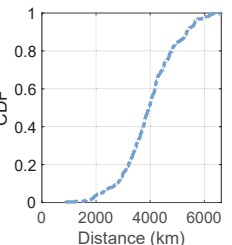
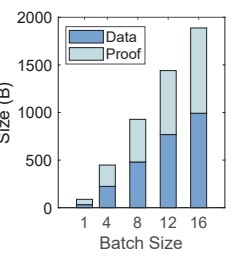

**(a) Multiple queried batches.** **(b) Improper storage node.** **(c) Large transmission size.**

**Figure 3: The root reasons for poor query performance.**

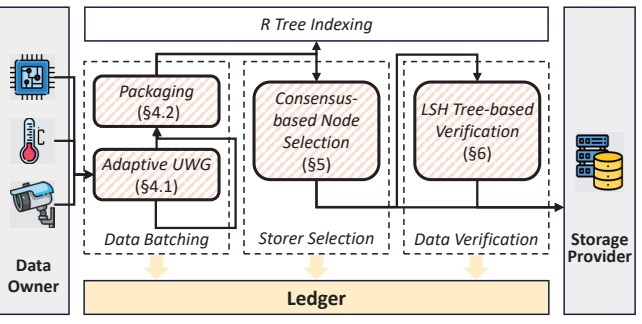

**Figure 4: Architecture of TimeChain.**

the existing dataset, YCSB [5]. As illustrated in Fig. 3a, more than 84.25% of queries span over 10 batches. When these batches reside on different nodes, additional query and transmission delays are introduced.

**2) Improper Storage Node Selection.** In this measurement, we found that the transfer latency accounts for a significant portion of the total query latency. As shown in Fig. 3b, there is a large difference in the distance of storage nodes around the world, which leads to a large difference in the transmission latency from one node to another. If a very far away storage node is selected, this leads to an increase in transmission latency. Moreover, in the presence of malicious nodes, the final choice of storage node may not be optimal in terms of transmission delay, which leads to additional transmission overhead.

**3) Large Size of Proof Transmitted.** To enable storage providers to quickly provide data owners with integrity proofs through flexible queries, proof data also needs to be stored with the storage providers. So when the storage provider needs to prove the integrity of the data to the data owner, the data proof is also sent back to the data owner. Fig. 3c shows the breakdown of the total transmitted data. We can find from the figure that the proof size occupied 48.8% of the received data, which is almost half of the received data. When the network is busy, a large size of proof will increase the network transmission delays.

## 3 TIMECHAIN OVERVIEW

In order to improve the query performance of the blockchain storage system, we design TimeChain, a novel blockchain-based storage system for IoT time series data. Fig. 4 shows the architecture of

TimeChain. TimeChain is built on the blockchain platform and all operations are recorded on the distributed ledger. The core modules in TimeChain include data batching, storer selection, and data verification. The indexing structure of TimeChain is R-tree, which can accelerate spatiotemporal aggregation searches commonly used in IoT scenarios. Next, we introduce the key modules of TimeChain.

**Data Batching Module:** Our measurement study reveals that improper packaging methods increase the number of queried spanning batches, thereby increasing network transmission delays. We construct an adaptive undirected weighted graph (UWG) to accurately capture user query information based on the historical query of data owners (§-4.1). For the UWG we built, the problem of what data to pack into batches is converted to a clustering problem [39], which divides all the original data into multiple clusters according to the user's query request. There are many traditional algorithms for solving clustering problems, such as K-means [21], GMM [16], etc. However, such traditional clustering algorithms are not suitable for dividing the data generated by IoT devices. This is because in TimeChain the user's query does not follow specific features, which may cause the clustering of the data graph to form a complex shape instead of the common circular shape. In addition, traditional clustering algorithms need to divide all data into a fixed number of sets, but not all sets are equal to the batch size, which will bring extra overhead to index queries. Therefore, we use the spectral clustering algorithm to pack data (§-4.2), which is very suitable for dealing with irregular and non-fixed numbers of clusters.

**Storer Selection Module:** The *selection* of storage nodes is crucial. Like we found previously in Fig 3b, the distance between the storage node and the client affects the data access latency. In addition, for off-chain storage databases, nodes with insufficient storage space or malicious nodes can cause data loss, tampering, or service interruption, which in turn affects the security and stability of the entire system. Therefore, we comprehensively evaluate storage nodes based on information such as distance and historical service records. The *security* of the storage node selection process is also very important. Storj [22], CoopEdge [41], and PipeEdge [40] select service nodes through a fixed set of nodes, and confirm the decisions by the blockchain. In other words, they make decisions centrally and still face the threat of single point failure [35]. However, using a voting mechanism similar to PBFT, the node selection process usually requires multiple rounds of task calculation and message broadcasting. And if the consensus process and the node selection process are completely decoupled, the system security will

be compromised. To solve this problem, we combine the node selection process with the consensus and propose a consensus-based node selection mechanism (§-5).

**Data Verification Module:** We can find from the previous measurement results, that close to half of the data transferred is data integrity proof. The data proof is organized as a Merkle tree, which is built from a series of hashing. In the Merkle tree, the number of hashes that are non-leaf nodes is almost equal to the number of original data points. Since the size of the IoT data units is approximately equal to the hash values, this means that the amount of data that needs to be sent to validate the data is almost twice as much as the original data. Reducing the size of data proof poses a challenge. Upon analyzing IoT data, we observe that IoT data changes slowly and rarely exhibits abrupt changes within a short period [17]. For these similar data, the Locality-Sensitive Hashing (LSH) algorithm can generate similar hash results from similar original data [20]. The LSH ensures that similar IoT data remain similar even after hashing. Therefore, we propose a novel LSH tree-based verification mechanism (§-6), which employs LSH instead of the universal hashing used in traditional Merkle trees. By differentially transmitting LSH hash values, the size of the transmitted data can be significantly reduced.

## 4 ADAPTIVE PACKAGING MECHANISM

For data packing, we construct an adaptive UWG based on historical queries to characterize the dynamic query range. By running the spectral clustering algorithm of the adaptive UWG, we pack the raw data into batches based on random user queries.

### 4.1 Adaptive UWG based on Historical Query

Since the raw data of IoT are isolated points, we create weighted edges between data points, which represent the probability of being jointly queried. The weight of the edge between points $a$ and $b$ is denoted as:

$$l_{ab} = \begin{cases} \sqrt{(id_a - id_b)^2 + (t_a - t_b)^2} & , k = 0 \\ \theta \cdot l_{ab} + (1 - \theta) \cdot x_{ab}^k & , k \geq 1 \end{cases} \quad (1)$$

where $l_{ab}$ is initialised to the Euclidean distance between the two device IDs and the time when no request arrives. When a user's request arrives, the UWG is dynamically adjusted according to the range of data involved in the request. To avoid excessive storage overhead of querying the graph, we ignore the time dimension of the data when updating the graph and only consider the device ID of the data. We use the flag variable $x_{ab}^k$ to indicate the content of the $k$th query. When the $k$th query contains device $d_a$ and device $d_b$, $x_{ab}^k = 1$, otherwise $x_{ab}^k = 0$. Then, the distance $l_{ab}$ will be updated according to $x_{ab}^k$. The data points in the graph cannot be predicted by a fixed pattern since the user requests may be very random. Therefore we set an influencing factor $\theta$ to determine the impact of the weight on the client's request. When $\theta$ is closer to 1, the weight is more affected by the query. When $\theta$ is close to 0, it means that the batch clustering is kept as initial as possible.

Through the adaptive weight clustering algorithm, we can dynamically adjust the weights between nodes according to the distance between nodes and the relevance of the query to better reflect

---

**Algorithm 1:** Packaging Algorithm

**Input:** $D, S, Q^i$.
**Output:** $P$.

1 **begin**
2    $S' \leftarrow \{s^a | a \in D \ \& \ s^a \subset S\}$
3    $L \leftarrow \left\{ \sqrt{(id_a - id_b)^2 + (t_a - t_b)^2} \Big| \exists_{a,b \in D} \right\}$
4    $X^k \leftarrow \{0 | \exists_{a,b \in D}\}$
5    **for** $q^k \in Q^i$ **do**
6      **if** $a, b \in q^k$ **then**
7        update $X^k$ with $x_{ab}^k \leftarrow 1$
8      **end**
9    **end**
10    $L \leftarrow \left\{ \theta \cdot l_{ab} + (1 - \theta) \cdot x_{ab}^k \Big| \exists_{a,b \in D} \exists_{l_{ab} \in L} \right\}$
11    $D' \leftarrow \text{cluster}(D, L)$
12    $P \leftarrow \{\}$
13    **for** $d^j \in D'$ **do**
14      add $\{s^a | \exists_{a \in d^j}\}$ to $P$
15    **end**
16 **end**
17 **return** $P$

---

their similarity. This helps to more accurately determine which nodes' data should be placed in the same batch during the packaging process to improve the efficiency and accuracy of the query.

### 4.2 Packaging Mechanism with Spectral Clustering Algorithm

Since the spectral clustering algorithm is suitable for handling classification problems with irregular shapes, we use it to pack the data. Algorithm 1 shows the total packaging process of the TimeChain. The input of the algorithm includes the input devices set $D$, data set $S$, and users' $i$th history query set $Q^i$. We first organize the original data $S$ into a set $S'$ according to the device name and time unit, which represents the data of a device in a time unit (line 2). We first initialize the weight set $L$ (line 3). For the historical query records $Q^{i-1}$ in the last interval, we collect the query information $X^k$ (lines 5-9). Then we update the weight set $L$ according to the collected query information $X^k$ (line 10). For the UWG $(D, L)$, we use the spectral clustering algorithm to obtain the aggregation result $D'$ (line 11). According to the aggregated result $D'$, we merge the data into $P$, which is the packaging result we get (lines 13-15).

## 5 NODE SELECTION MECHANISM BASED ON CONSENSUS PROTOCOL

### 5.1 Protocol Process

In order to safely select optimal storage nodes, TimeChain proposes a node selection algorithm based on a consensus mechanism. The total selection process includes *request*, *prepare*, *pre-submit*, *submit*, *submit*, and *reply*, the details of which are shown in Fig. 5. Similar to PBFT consensus, in the *request* phase the gateway sends a request to all nodes in the system, and the consensus nodes will return the

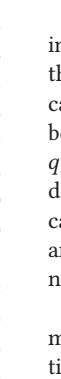

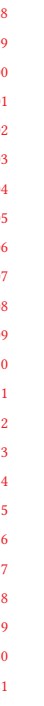

**Figure 5: Workflow of consensus-based node selection.**

obtained results to the gateway in the *reply* phase. Again similar to PBFT consensus, we suppose the number of Byzantine satellites is $f$, and the number of total satellites is more than $3f + 1$.

In the *prepare* phase, each node calculates the score by considering the distance, reputation, etc. of the storage node and broadcasts the score to all other nodes. We use $p_i = \alpha \cdot d_i + \beta \cdot s_i + \gamma \cdot q_i$ to calculate the score of the $i$th node, where $d_i$ denotes the distance between the $i$th node and the client node, the storage service quality $q_i$ can be evaluated from the service records on the chain, and $s_i$ denotes the remaining storage space of the node. All these data can be found on the chain. $\alpha$, $\beta$, and $\gamma$ are weighting parameters, and these coefficients can be adjusted according to specific system needs and performance requirements.

In the *pre-commit* stage, the consensus node receives prepared messages set $\{p_i\}$ from other nodes. When the timer of this node times out and more than $2f + 1$ prepare messages are received, consensus nodes decide the optimal storage provider according to the reputation priorities $\{p_i\}$ they receive. If each round of consensus only returns the closest node, due to the influence of distance on the reputation calculation mechanism, the storage pressure on some closer nodes may be very high. To balance the load, the consensus nodes will return at a set of the highest reputation $n$ nodes for gateway to randomly select, instead of the highest reputation node.

In the *commit* stage, all nodes will receive the optimal storage decision recommended by other nodes. When the number of the same pre-commit messages exceeds $f + 1$, this node will commit the optimal storage node to the client.

## 5.2 Security Analysis

We consider here the security of this consensus protocol. Since TimeChain's consensus protocol just adds extra information based on PBFT, we only consider the security risk posed by the extra information in the *prepare* and *pre-commit* phases. In the *prepare* phase, if a node fakes its own score, the authenticity of the score can be easily checked by the gateway since the evaluation data sources can all be found on the chain. Once a node falsifies its reputation, the behavior will also be recorded on the chain, thus affecting the next reputation assessment. Moreover, since only one storage node is selected at the end, gateway does not pay attention to the authenticity of scores of all nodes, but only the score of the selected node. In the *pre-commit* phase, if any node forges the final score, it does not affect the final result. This is because for a network

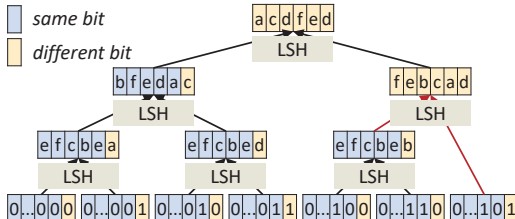

**Figure 6: LSH tree.**

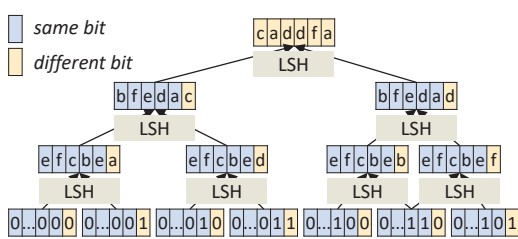

**Figure 7: A non-full binary LSH tree with tail merging.**

of $3f + 1$ nodes containing $f$ Byzantine nodes, $f + 1$ identical results must be obtained in the *commit* phase.

## 6 LSH TREE-BASED VERIFICATION MECHANISM

To address the high network transmission delay caused by the large amount of verification data transmitted, we propose a novel LSH tree and optimize its space through a tail merging strategy.

### 6.1 LSH Tree

Locality-sensitive hashing maps similar data to similar hash values for deduplication. In TimeChain, we usually package data that is close in physical space or time, which tends to have high local similarity. Therefore, by using locality-sensitive hashing on the same batch, we can get similar hash values. When a hash value needs to be transmitted, only the hash difference part is transmitted, thereby reducing the amount of data to be transmitted.

We show an example of an LSH tree in Fig. 6. Specifically, for the data in a batch, we take steps similar to the Merkle Tree, first performing local-sensitive hashing on the original data. For the hashed result, we merge two close hashes into a string and calculate the locality-sensitive hash value of the string. We then recurse this process upwards, layer by layer, until we get a unique hash, the hash root. In the first level of hashing, many bits of hash value are the same due to the high similarity of the original data. Therefore, when transmitting the hash value of the first level, we can only transmit different bits to reduce the transmission delay. Likewise, since there are local similarities in the first-level hash, many bits in the second-level hash will also be similar. By analogy, for the hash value of each layer, we only need to transmit the hash difference bits, thus further reducing the amount of data transmitted.

Since we are using LSH tree instead of the original Merkle tree, we need to analyse the security of LSH tree. Here we mainly consider the scenario where the storage provider tampers with the

data, i.e., the case where the same hash can still be obtained with the different original data. We tested for different layers of the LSH tree and found that in the layer of hash closest to the data source, the hash has an average difference bit count of 170bit. This has surpassed the MD5 and SHA1 standards, which are now very commonly used in IoT scenarios.[11, 24]. For the layers of the LSH tree close to the root node, despite the smaller number of hash difference bits, this does not make sense for tampering with the original data.

## 6.2 Tail Merging

In a full binary tree, LSH Tree can perform locality-sensitive hashing by merging data in batches in pairs. However, if the number of data in the batch is not sufficient to form a full binary tree, building a hash tree like a Merkle tree will result in a loss of similarity. For example, in Fig. 6, there are 7 data points in a batch, which size is not satisfied for a full binary tree. In the first round of hashing, the first 6 data perform LSH in pairs. The hash results of these 6 data are similar, due to the similarity of the raw data. In the second round of hashing, because the first round of hash results of data 5-6 and the 7th raw data are very different, the hash results of these two are also very different from the hash results of data points 1-4. When performing integrity proofs, all of these dissimilar hashed data bits need to be transmitted, which increases the amount of data transferred.

In order to solve this problem, we introduced the tail merging strategy, which merges the tail nodes of the non-full binary tree with the front nodes of the same layer. As shown in Fig. 7, in the first round of hashing, we merge the left out the 7th node with the the 6th node, in order to preserve the similarity of the data as much as possible. The hash result of the data 6-7 is efcbeb, and the hash value of the data 5-6 is efcbef. Obviously, there exists high similarity in the first round of hashing, and it can be maintained to the next level of hashing. This reduces the transmitted hash value size from 12 bits to 7bits, at the cost of only transmitting 1 more different bit in the first round of hashing. In this way, when doing integrity proof, we only need to transmit the different hash bits, thus reducing the amount of data transmitted.

## 7 EVALUATION

In this section, we evaluate the storage performance and query performance of TimeChain.

## 7.1 Experimental Setup

We implement TimeChain based on top of some open-source projects, such as Hyperledger Fabric and IPFS. The block size is set to 1500 and the block interval is 1 second. We simulated 320 cloud server nodes distributed in different locations around the world and conducted experiments based on this cluster. Each storage node is configured with a 2-core CPU and 4GB memory, the storage space of each storage node is 512GB. The distance between the storage node and the gateway ranged from 800km to 6000km, with an average of 4000km. Considering that some storage providers are fraudulent, the data stored remotely will be inaccessible with a probability of 60%. We use a PC as the gateway node of the IoT sensors, which is equipped with Intel(R) Core i7-13700K CPU @

5.4GHz, 32GB DRAM, and runs Ubuntu 22.04. The default batch size and query size are set to 20.

*7.1.1 Baselines.* **SEBDB** [46] is a typical representative of the on-chain databases. It enables efficient access to on-chain blocks by storing all data on the blockchain and using the B+ tree to create a fast index on timestamps and device names. In terms of data verification, SEBDB uses the traditional Merkle tree for data verification. Merkle tree achieves data integrity verification by calculating hash values of data blocks and organizing these hash values into a tree structure layer by layer.

**FileDES** [38] is a file-based storage system. It achieves safe storage and reliability of data by storing data on remote nodes and recording the hash value of the data on the chain. When a client needs to search for data, FileDES traverses all blocks on the blockchain to get where the data is stored. In terms of data verification, FileDES also uses the same Merkle tree as SEBDB.

*7.1.2 Dataset and Workloads.* We use the following three datasets: Hong Kong–Zhuhai–Macao Bridge (Bridge) [44], RT-IFTTT (RT) [17] and Weather (WX) [1]. Considering the storage characteristics of the time series storage system [28], we append the device information in the dataset to the header of the sensor values, which takes up 56B of space per piece of data. We set the average data query range of these three datasets as 100, 20 and 10 respectively based on the data generation rate of these three datasets.

## 7.2 Overall Performance

**Storage Latency.** We compared the storage latency under different batch sizes. Due to the large number of IoT devices and the fast data generation speed, we define storage latency as the total latency for storing 10,000 data, rather than focusing on the micro latency of a single data. As shown in Fig. 8, TimeChain has lower storage latency than both SEBDB and FileDES for different batch sizes. This is because TimeChain 's unique packing mechanism and node selection mechanism reduce the data transfer latency. Moreover, as the batch size becomes larger, the storage latency becomes smaller. This is because, for the same amount of data, when the batch size is larger, the number of times the data is packed and recorded in the chain decreases. Users can control the storage latency by adjusting the batch size.

**Query Latency.** We compare the query latency with different batch sizes, which is shown in Fig. 9. The query latency refers to the average latency of randomly querying a data set based on a fixed query range. From Fig. 9, we can see that the query latency of TimeChain is lower than the other two schemes. This comes from TimeChain 's reduction in data transfer latency for reasons that will be explained in Fig. 10. We can also discover that when the batch size increases, the query latency decreases. This is because when the batch size increases, the number of batches involved in the same query decreases, and the user's query results will try to concentrate on one storage node. However, the larger the batch size, the improvement of TimeChain over other solutions will diminish as the batch size increases. This is because when the batch size is very large, it is equivalent to storing all the data in a single batch, in

---

[1] https://www.kaggle.com/selfishgene/historical-hourly-weather-data/

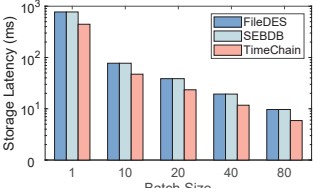

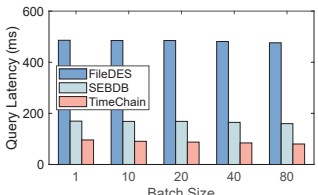

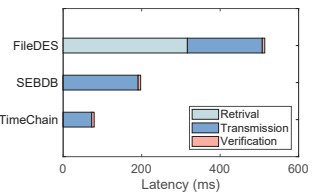

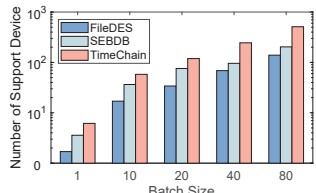

**Figure 8: Storage latency.**      **Figure 9: Query latency.**      **Figure 10: Breakdown of query latency.**      **Figure 11: Numbers of supporting device.**

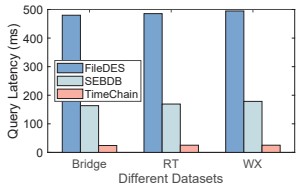

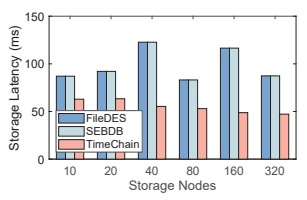

**Figure 12: Query latency under different query size.**      **Figure 13: Storage latency under different network.**

which case data clustering does not lead to performance improvement. Moreover, when a large amount of data is concentrated in a storage node, the scalability and reliability of the storage system will also be damaged.

**Breakdown of Query Latency.** We further analyze the breakdown of query latency, which is shown in Fig. 10. The latency of a query is mainly composed of 4 stages, retrieval, transmission, verification, and return. Considering that sensors usually choose a closer gateway, the latency of the return stage can be almost ignored. In the validation phase, the latency of the three schemes is relatively close to each other, which is less than 1 ms and also almost negligible. The delays in the transmission and retrieval phases account for the major part of the query delay. The transmission delay of TimeChain is significantly lower than that of FileDES and SEBDB. This is because of TimeChain's unique node packaging mechanism and selection mechanism, which reduce the number of data fetching times and shorten the distance from storage providers. For the retrieval stage, since FileDES traverses all blocks to retrieve data, the retrieval delay is particularly high. While SEBDB and TimeChain respectively use B+ and R trees to build indexes respectively, thus reducing the latency to less than 1ms.

**Maximum Number of Storage Devices Supported.** Specifically, we use the metric of maximum number of supported devices, which refers to the number of devices that the storage system can support for storage services per second. We assume that a gateway can handle data storage requests from multiple IoT devices, and ignore the processing delay of the gateway itself. All IoT devices simultaneously generate data at 1hz and require that this data must be stored before the next data is generated. As shown in Fig. 11, TimeChain increases the maximum number of supported devices by 1.63x and 3.55x compared to SEBDB and FileDES, respectively. This is mainly due to the fast storage latency of TimeChain, where data transfer latency is very low and allows TimeChain to store data at a much faster rate. Moreover, the maximum number of supported devices will increase as the batch size increases. When the

batch size is up to 80, the maximum number of devices supported by TimeChain has reached thousands.

### 7.3 Performance under Different Parameters.

**Query Latency under Different Query Size.** We compare the query performance on three different query size datasets: Bridge, RT, and WX, the result of which is shown in Fig. 12. The average query size of the workloads in the three data sets is 10, 20, and 40 respectively. We can find that the query latency of these three solutions usually decreases with the increase of query size. This can be attributed to the fact that a larger query size means more data is covered in the query, thereby improving data locality and query efficiency. We observed that when the query size becomes larger, the performance improvement brought by TimeChain also increases, compared to SEBDB and FileDES. This is because when the query size becomes larger, SEBDB and FileDES often need to obtain data from more nodes than TimeChain.

**Storage Latency Under Storage Network Scale.** We compared the storage latency of each solution under different storage network scales, as shown in Fig. 13. As the number of storage nodes increases, the storage latency of TimeChain shows a downward trend. This is because when the number of nodes increases, gateways in TimeChain can choose more storage providers, which increases the probability that closer nodes will be chosen. Other solutions will not benefit from the increase in the number of storage nodes. This is because both FileDES and SEBDB randomly select storage nodes, and the growing storage nodes will not significantly affect the results of random selection. Therefore, the storage latency of FileDES and SEBDB shows a relatively large randomness.

### 7.4 Ablation Study

In this subsection, we demonstrate the improvement effect of our design through three ablation studies.

**Clustering Algorithm.** We compared the effect of clustering algorithms and compared TimeChain with SEBDB. As shown in Fig. 14a, the network transmission delay of TimeChain is reduced by 40.3% compared with SEBDB. The packaging nodes in SEBDB do not consider the regularity of user queries, and the aggregated data is not divided according to the type of data source. When the data owner requests a series of data, the SEBDB gateway needs to cross multiple batches from more storage providers. TimeChain uses a spectral clustering algorithm to package data from specific sensors based on the characteristics of user requests, resulting in 59.3% fewer access batches than SEBDB, as shown in Fig. 14b.

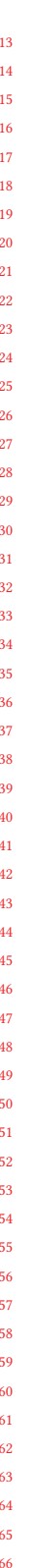

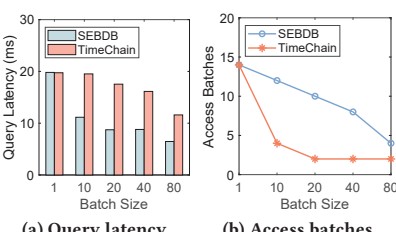

(a) Query latency.    (b) Access batches.

**Figure 14: Clustering ablation study.**

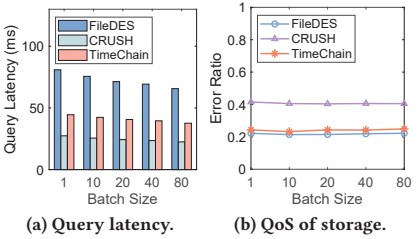

(a) Query latency.    (b) QoS of storage.

**Figure 15: Node selection ablation study.**

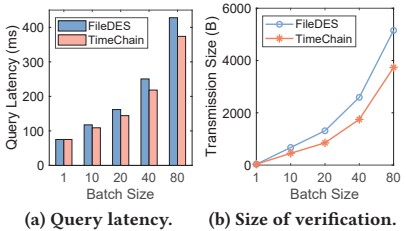

(a) Query latency.    (b) Size of verification.

**Figure 16: LSH tree ablation study.**

**Node Selection.** We show the performance differences between TimeChain, FileDES, and CRUSH in node selection in Fig. 15a. FileDES [38] divides some trusted storage nodes based on node reputation and randomly selects nodes from the set of nodes to store data. Therefore FileDES has the highest probability of storage service provision among the three, as shown in Fig. 15b. However, the random node selection of FileDES may introduce more distant storage nodes and thus longer transmission delays. CRUSH [34] selects the nearest node for storage based on the physical location of the node, but does not take into account storage node failures and single points of failure. This makes it possible for CRUSH to select closer but unreliable storage nodes, so that 41% of the nodes selected by CRUSH are unable to provide effective storage services. TimeChain, on the other hand, takes into account the physical distance of nodes and node reputation and has the best performance both in response time and serve probability. Although the node distance selected by TimeChain is not the closest, TimeChain works best considering the node distance and quality of service.

**LSH Tree.** We compare the network transmission latency of FileDES and TimeChain as shown in Fig. 16a. We can find a 10.9% reduction in data transfer latency for TimeChain compared to FileDES. This is due to the fact that when there are more sensor devices in the local area network and the frequency of data generation is high, the amount of data transmission significantly affects the transmission delay in a congested network. As shown in Fig. 16b, the amount of data transmitted over the network is greatly reduced due to the fact that TimeChain uses LSH as hash algorithm. This will reduce the storage and query latency and greatly reduce the storage burden on the storage provider.

## 8 RELATED WORK

**Distributed Storage Systems.** In recent years, distributed databases have gained increasing prominence due to the inherent vulnerabilities of centralized data storage solutions, particularly their susceptibility to single points of failure. Apache Cassandra [23], Spanner [12] and CockroachDB [31] are all distributed key-value storage systems that ensure high availability and fault tolerance of data storage through special data distribution strategies. However, these distributed storage systems still suffer from single-point failure and high storage costs. Even when data has been distributed to multiple storage nodes, control of the data remains in the hands of the database vendor, making it easy for storage system companies to tamper with the data without the data owner's knowledge.

**Blockchain Storage Systems.** These days, many studies have focused on the performance of on-chain data storage, including query performance and storage burden. SEBDB [46] and MSTDB [45] do this by introducing different indexing mechanisms in order to support various types of queries such as SQL-like queries and semantic-based multi-keyword queries. Different from establishing a block-level index, LVMT [25] and COLE [43] optimized the index on the Merkle Patricia Trie to improve the query speed for the blockchain state. For the burden of storing data in the ledger, Rapidchain [42], SlimChain [37] and GriDB [18] distribute the ledger to other shards for storage to reduce the storage pressure. TimeChain can easily be combined with these on-chain storage optimisation solutions to accelerate the acquisition of on-chain hashes. However, in IoT scenarios with large data volume and fast generation, these solutions not only bring the risk of data privacy leakage, but also bring additional space storage burden to the blockchain, which is unsuitable for IoT scenarios with fast data generation.

**Blockchain-based File Systems.** Blockchain-based file systems have received extensive research and attention, due to their assurance of file integrity. Filecoin [6] is a decentralised file storage system built on IPFS [7], which encourages users to provide storage services by means of an incentive mechanism. Storj [22] and Sia [32] establish a Merkle tree for each file in a semi-decentralized way to ensure file integrity. FileDES [38] focuses on encrypted storage of data. It protects the security of data storage by introducing technologies such as zero-knowledge proof. However, none of these methods can provide efficient IoT data storage. Because the low value density of IoT data, if a single data is stored in the form of a file in the blockchain file system, it will bring very high costs.

## 9 CONCLUSION

In order to integrate IoT data with fast data generation speed and large data volume into blockchain with slow transaction processing speed for security, we propose TimeChain, an efficient off-chain blockchain storage system for time series data, which uses a batch processing method for discrete data. TimeChainpackages IoT data onto the chain, reducing storage latency. We propose an adaptive packaging mechanism, node selection mechanism based on consensus protocol and LSH tree-based verification mechanism to improve the query performance of TimeChain. We have implemented the system based on an open-source framework. Experiments have shown that compared to existing work, name can reduce query latency by an average of 64.6% and storage latency by 35.3%.

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
