# OpenReview forum: "TimeChain: A Secure and Decentralized Off-chain Storage System for IoT Time Series Data"
_ACM.org/TheWebConf/2025/Conference — WWW 2025 Poster_

### Official Review · Reviewer_2qjB · 2024-11-04

**Novelty:** 2
**Technical Quality:** 3

**Review:**

The paper presents TimeChain, a blockchain-based off-chain storage system for IoT time series data.

TimeChain stores only the hash of data batches on-chain while keeping full data off-chain to minimize storage costs and improve system performance. Key features include an adaptive batching mechanism, graph partitioning for optimized co-querying, and a Locality-Sensitive Hashing (LSH) integrity verification approach to reduce latency.


Positive side:

- The paper proposes a solution by integrating adaptive data batching, node selection mechanisms, and LSH-based verification to improve off-chain storage for IoT time series data. This combination is relevant for addressing latency and data management in blockchain-based storage.

- The authors clearly articulate the challenges faced by current blockchain-based storage systems, particularly in handling IoT data.

- The paper includes an in-depth evaluation with real-world datasets and comparisons to existing solutions (e.g., SEBDB and FileDES), showcasing improvements in both storage and query latency.

- The solution accounts for scalability by optimizing data batching and minimizing transmission size through LSH-based deduplication, making it suitable for high-volume IoT data.

- Overall presentation is good.

Negative side:

Novelty:

Storing a hash or tag on-chain as an identifier is a well-explored and intuitive approach, which does not contribute significantly to the novelty of this work. The main value seems to be derived from the actual off-chain processing and mechanisms. A primary concern with this paper is how effectively it highlights genuine research questions. Currently, it feels like the work represents an incremental step rather than a major advancement. Additionally, this study might be more appropriate for a journal publication (e.g., IEEE Transactions on x) rather than a conference like WWW.

Technical Details:

- The proposed consensus-based node selection mechanism is permissioned, implying that the nodes during specific periods need to be fixed and static. It is unclear how the system adapts to real IoT scenarios where nodes frequently enter sleep mode or become inactive.

- This also raises an additional concern regarding scalability. It remains unknown how well this consensus-based node selection scales as the number of nodes in the network significantly increases.

- There should be a deep theoretical analysis of the complexity involved in using LSH trees, along with a discussion on why other tree structures were not considered. This could include a detailed comparison of construction costs and performance across various data types.

- The paper's focus on specific IoT datasets and scenarios limits its applicability to broader use cases. There is minimal discussion about potential challenges or considerations when deploying TimeChain in different blockchain networks or diverse data environments.

**Questions:**

N/A

**Reviewer Confidence:**

4: The reviewer is certain that the evaluation is correct and very familiar with the relevant literature

**Scope:**

3: The work is somewhat relevant to the Web and to the track, and is of narrow interest to a sub-community

---

### Official Review · Reviewer_bxv9 · 2024-11-28

**Novelty:** 4
**Technical Quality:** 5

**Review:**

The paper demonstrates high technical quality, presenting a well-structured, comprehensive, and rigorous evaluation of TimeChain, a blockchain-based off-chain storage system for IoT time series data. The implementation of the system using open-source frameworks (e.g., Hyperledger Fabric and IPFS) and the use of real-world datasets highlight the practicality and feasibility of the solution. The experiments are thorough, covering a wide range of metrics (e.g., storage latency, query latency, and reliability) and baselines (FileDES and SEBDB).

The paper is generally clear and easy to follow, with a logical flow from problem definition to solution design and evaluation.

The work is original and addresses a critical challenge in IoT and blockchain integration: efficiently storing and querying high-frequency, low-value-density IoT data.

The significance of this work lies in its practical implications for IoT and blockchain ecosystems.

Pros:
- Integrate the spectral clustering algorithm with data batching technique for efficient query
- Introduce a node selection mechanism combining distance, storage capacity, and reputation into a consensus-based framework.
- Propose a LSH tree-based verification reduces data transmission overhead during integrity checks.

Cons:
- The experimental section lacks detailed configuration and description of the blockchain consensus node.
- The weighting formula of the node selection mechanism may not be able to cope with the complex network environment. It is suggested to discuss in depth, including but not limited to: how to adjust the weighting parameters, the rationality of selecting these three indicators, etc.

**Questions:**

- Is PBFT suitable for this kind of large-scale data volume scenario?
- The data generation of the Internet of Things is very fast, and the PBFT algorithm itself will become a bottleneck. Have you considered using other consensus algorithms?

**Reviewer Confidence:**

2: The reviewer is willing to defend the evaluation, but it is likely that the reviewer did not understand parts of the paper

**Scope:**

4: The work is relevant to the Web and to the track, and is of broad interest to the community

---

### Official Review · Reviewer_LVq1 · 2024-11-29

**Novelty:** 3
**Technical Quality:** 4

**Review:**

### Quality
To some extent, this paper shows technical quality by addressing the challenges of storing and querying IoT time-series data using blockchain technology. It further illustrates the effectiveness of the proposed methodology through comprehensive experiments.

### Clarity
The paper’s writing is unclear and lacks clarity. For instance, several technical terms (e.g., t_a, t_b) are introduced without proper definitions in Section 4. Additionally, there are typographical errors, such as “TimeChainpackages” and “name” in the conclusion. Furthermore, Figure 14(a) shows the query latency for TimeChain as higher than SEBDB, which contradicts the explanations in the text.

### Originality
To some extent, the paper shows novelty, which introduces a new approach to address the efficiency problem of IoT storage based on blockchain. This paper incorporates blockchain-related technologies, such as off-chain storage, and proposes new technologies, such as adaptive packaging and LSH tree.

### Significance
The paper provides a practical solution for efficiently handling the high volume of low-value-density IoT data securely and decentralized. The paper demonstrates substantial improvements in query latency and storage efficiency compared to existing systems.
However, it falls short in discussing real-world deployment challenges, such as handling heterogeneous IoT environments and regulatory constraints. The security analysis of the LSH tree is not well-illustrated, especially its resilience to adversarial attacks.

### Pros
1. The motivation of this paper is clear.
2. This paper introduces a new approach to effectively address the IoT time-series data store and query problem, showin improvements in storage and query latency.
3. This paper uses spectral clustering for adaptive data packaging, which improves query performance by reducing the number of nodes accessed during queries. This demonstrates scalability in data grouping and clustering.

### Cons
1. The writing of this paper is poor.
2. In Sections 4 and 5, the analysis of the proposed mechanisms lacks details about trade-offs and potential limitations, such as coefficient adjustment, communication overhead, and applicability to various deployment conditions.
3. Several concepts and definitions need more clarity and academic rigor. For instance, Section 2.1 does not explicitly state whether the proposed structure of the basic blockchain-based distributed storage system is intended as a general model applicable to all scenarios or is specifically tailored for IoT. Additionally, the introduction of the term "satellite" appears to be specific to the IPFS; however, it is presented without adequate contextualization or definition.

**Questions:**

1. Please clarify the rationale behind selecting key parameters, such as batch size and the weights used in the node selection mechanism. Additionally, could you elaborate on specific technical considerations, such as the trade-offs in applying spectral clustering for adaptive packaging?
2. Since the clustering mechanism is based on historical queries, how does TimeChain adapt if there are abrupt changes in query patterns? Would the system require retraining, or is it flexible enough to handle such changes dynamically?
3. Could you add the selection rules of the baselines? For example, why did you choose FileDES to be compared? Is it a well-known or well-used system? Please explain the rationale behind selecting the baselines and the importance or practicability of the baselines.
4. The introduction highlights the advantages of off-chain over on-chain storage for IoT data scenarios, and this analysis led to the design of the off-chain TimeChain system. However, why was SEBDB, an on-chain method, chosen as the primary baseline for evaluation? Furthermore, since CRUSH also utilizes physical node locations, why was its comparison limited to ablation studies rather than a more comprehensive evaluation against TimeChain?
5. Could you provide more details about the measurement study setting in Section 2.2? Specifically, how were the 300 simulated storage providers implemented, and does this simulation accurately reflect real-world conditions?
6. Please provide more details on the consensus-based node selection. Since PBFT requires message broadcasting, how does the protocol achieve high performance in a high-latency real-world environment? The consensus protocol of TimeChain is just based on PBFT with extra information, how to show its novelty?

**Reviewer Confidence:**

2: The reviewer is willing to defend the evaluation, but it is likely that the reviewer did not understand parts of the paper

**Scope:**

3: The work is somewhat relevant to the Web and to the track, and is of narrow interest to a sub-community

---

### Official Review · Reviewer_fUr4 · 2024-12-02

**Novelty:** 4
**Technical Quality:** 4

**Review:**

This paper introduces TimeChain, an efficient off-chain blockchain storage system designed for time series data. It leverages a batch processing approach for handling discrete data. Additionally, the authors propose an adaptive packaging mechanism, a consensus-protocol-based node selection strategy, and an LSH tree-based verification method to enhance TimeChain's query performance.

**Questions:**

Thank you to the authors for presenting this interesting paper. Below are a few comments:

1) Blockchain-based distributed storage systems are often plagued by scalability issues. However, the paper appears to overlook the potential scalability challenges in TimeChain. Without addressing these concerns, it may be difficult for readers to believe that TimeChain can be effectively deployed in real-world scenarios.

2) The paper asserts that TimeChain is secure against attacks, but the evaluation section does not provide experimental evidence to substantiate this claim. Blockchain-based systems are known to be susceptible to attacks such as 51% attacks or majority attacks, where a single entity could dominate the network's computational power. Furthermore, if the system depends on smart contracts, vulnerabilities in the contracts could undermine its security. The authors should clearly elaborate on how TimeChain addresses these potential security risks.

3) The "Security Analysis" section is difficult to follow, and the connection between the security analysis and the experimental results is unclear. This part would benefit from a clearer explanation of how the security analysis ties into the presented experimental findings.

**Reviewer Confidence:**

3: The reviewer is confident but not certain that the evaluation is correct

**Scope:**

3: The work is somewhat relevant to the Web and to the track, and is of narrow interest to a sub-community

---

### Official Review · Reviewer_ppka · 2024-12-03

**Novelty:** 6
**Technical Quality:** 6

**Review:**

This paper proposes TimeChain, an efficient off-chain blockchain storage system for IoT time series data, which batches discrete time series data, storing only the hash value of each batch on-chain while keeping the complete data off-chain. The storage overhead and latency on the blockchain are significantly reduced by up to 37 times, which is very impressive. The TimeChain performs well in the experimental study.


Strong points:
1: Through measurement, it was identified that the bottlenecks of off-chain storage lie in the packaging mechanism, node selection mechanism, and verification mechanism. In response, Timechain proposes a DAG-based packaging mechanism, a consensus protocol-based node selection mechanism, and a locality-sensitive hashing tree-based verification mechanism.
2: Effective experimental results: TimeChain conduct experiments based on 50 real nodes. Compare to existing distributed architectures, TimeChain reduce query latency by 64% and storage latency by 35%.
3. The paper writing is good and easy to follow.


Weak points:
1: Insignificant Advantage Over Other Distributed Storage for Blockchain-based Data Storage: While decentralization can indeed enhance system robustness, it also introduces more consistency requirements and storage overhead, making its advantage over other distributed storage systems less pronounced.
2: Deficiencies in Security Risk Analysis:
a) Compared to traditional Merkle trees, locality-sensitive hashing trees, while reducing the amount of data for verification, may also introduce additional security risks and computational overhead. These potential issues require thorough analysis during the design and implementation process.
b) The node selection mechanism based on consensus protocols differs from the PBFT protocol in terms of broadcast information, necessitating further in-depth analysis of the impact of this difference on the security of the consensus process.
3: The experiments conducted for TimeChain are all based on a testbed, lacking real-user feedback.

**Questions:**

1. Is it possible to provide some user feedback about timechain?
2. Can you provide some analyses of consistency issues of TimeChain?

**Reviewer Confidence:**

4: The reviewer is certain that the evaluation is correct and very familiar with the relevant literature

**Scope:**

4: The work is relevant to the Web and to the track, and is of broad interest to the community